# Adaptive Masked Weight Imprinting for Few-Shot Segmentation

**Mennatullah Siam**
University of Alberta
mennatul@ualberta.ca

**Boris Oreshkin**
Element AI
boris@elementai.com

## Abstract

Deep learning has mainly thrived by training on large-scale datasets. However, for continual learning in applications such as robotics, it is critical to incrementally update its model in a sample efficient manner. We propose a novel method that constructs the new class weights from few labelled samples in the support set, while updating the previously learned classes. Inspiring from the work on adaptive correlation filters, an adaptive masked imprinted weights method is proposed. It utilizes a masked average pooling layer on the output embeddings and acts as a positive proxy for that class. It is then used to adaptively update the 1x1 convolutional filters that are responsible for the final classification. Our proposed method is evaluated on PASCAL-$5^i$ dataset and outperforms the state of the art in the 5-shot semantic segmentation. Unlike previous methods, our proposed approach does not require a second branch to estimate parameters or prototypes, and it enables the adaptation of previously learned weights. We further propose a novel setup for evaluating incremental object segmentation which we term as incremental PASCAL (iPASCAL), where our adaptation method has shown to outperform the baseline method.

## 1 Introduction

Few-shot learning literature has mainly focused on image classification Finn et al. (2017)Koch et al. (2015)Lin et al. (2017)Vinyals et al. (2016)Snell et al. (2017) Qi et al. (2017)Qiao et al. (2017). However, unlike image classification, semantic segmentation requires to learn a pixel-wise classification and can provide multiple classes in the support set. Thus, segmentation is more challenging in the low-shot regime than classification. One of the methods that are based on learning a parameter predictor was proposed by Shaban et al. (2017). A conditional network method was proposed by Rakelly et al. (2018) based on sparse or dense labels to guide the segmentation network. Another method that inspires from prototypical networks was proposed by Dong & Xing (2018). The previous methods require the training of an additional branch to act as a prototype learner or a parameter prediction branch. If a continuous stream of data is presented to the model that has annotations for both novel and previously learned classes, there is no direct extension to adapt their model. A concurrent work to ours is proposed by Zhang et al. (2018) that proposes a similarity guided network similar to Rakelly et al. (2018) but it utilize only one branch.

In this paper we propose an adaptive masked weight imprinting scheme for few-shot semantic segmentation. Our main inspiration is from classical approaches in learning adaptive correlation filters Bolme et al. (2010) Henriques et al. (2015). Correlation filters date to 1980s by Hester & Casasent (1980) that proposed learning an averaged matched spatial filter constructed as a weighted linear combination of basis functions. Bolme et al. (2010) proposed a fast object tracking method based on adaptive correlation filters, where the filters are updated using a running average. Our method proposes a novel scheme to compute convolutional filters to match the objects through masked weight imprinting, while adapting the learned ones. Weight imprinting Qi et al. (2017) has been proposed for image classification and relates metric learning methods to softmax classification. It utilizes the normalized embeddings for the support set as proxies and concatenate it to the original weight matrix in the last classification layer. However, the method does not provide a way to adapt the weights that were previously learned which would not be suitable for a continuous data stream. It

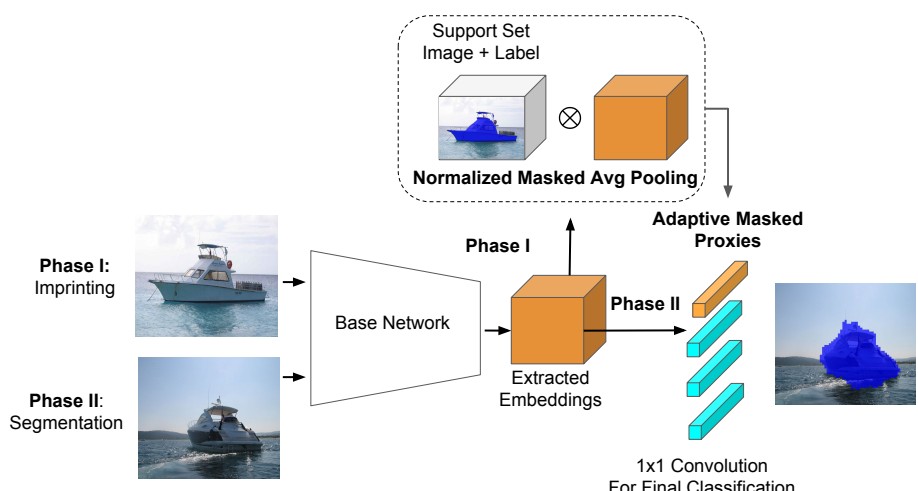

Figure 1: Adaptive Masked weight Imprinting method.

was designed mainly for classification and not semantic segmentation which is more challenging to operate with few labelled samples.

Unlike previous methods, our approach can easily operate with any pretrained network without the need to train a second branch. The contributions of this paper are: (1) we propose a multi-resolution masked weight imprinting scheme for few-shot segmentation. (2) we propose a novel adaptive weight imprinting scheme that inspires from adaptive correlation filters, in order to update the weights of previously learned classes. (3) Our method outperforms the state of the art on the 5-shot case on PASCAL-$5^i$, and the adaptation method outperforms the baseline method. (4) We propose iPASCAL which is the incremental version of PASCAL-VOC to evaluate incremental object segmentation.

## 2 PROPOSED METHOD

We formulate a problem similar to Shaban et al. (2017) for the few-shot setting. However, for the incremental object segmentation setup we propose a novel setup using incremental PASCAL which we term as iPASCAL. The PASCAL VOC dataset Everingham et al. (2015) classes are split into $L_{train}$ and $L_{incremental}$ with 10 classes each, where $L_{train} \cap L_{incremental} = \emptyset$. The classes belonging to the $L_{train}$ are used to construct the training dataset $D_{train}$ and pre-train the segmentation network. Unlike the static setting in the few shot case, the incremental mode provides the image-label pairs incrementally with different encountered tasks. Each task introduces two novel classes to learn. The tasks are in the form of triplets $(t_i, (X_i, Y_i))$, where $(X_i, Y_i)$ represent the overall batch of images and labels from task $t_i$. The batch labels are for the two novel classes belonging to task $t_i$, and the previously learned classes in the encountered tasks $t_0, ..., t_{i-1}$. In each task the model encounters each image-label pair from the current batch only once.

The backbone architecture used in our segmentation network is a VGG-16 Simonyan & Zisserman (2014) that is pre-trained on ImageNet Deng et al. (2009). Inspiring from the work in few-shot image classification with imprinted weights Qi et al. (2017) we propose to utilize a masked weight imprinting scheme. We utilize the embeddings for the few labelled samples from the novel class as proxies. The weight filter is used as a proxy that convolves the output feature map and computes the extent to which the different parts in the feature map matches these proxies. In order to incorporate the pixels that belong mainly to the novel class, masked feature maps with the binary labels provided

in the support set are used. This is followed by average pooling the masked feature maps per channel as in equation 1, we denote this layer as masked average pooling Zhang et al. (2018).

$$P_l^r = \frac{1}{k} \sum_{i=1}^{k} \frac{1}{N} \sum_{x \in X} F^{ri}(x) Y_l^i(x) \tag{1a}$$

$$\hat{P}_l^r = \frac{P_l^r}{\|P_l^r\|_2} \tag{1b}$$

Here $Y_l^i$ is a binary mask for $i^{th}$ image with the novel class $l$, $F^{ri}$ are the corresponding output feature maps for $i^{th}$ image and $r^{th}$ resolution. $X$ is the set of all possible spatial locations and $N$ is the number of pixels that are labelled as foreground for class $l$. The normalized output from the masked average pooling layer $\hat{P}_l^r$ can be further used as proxies representing class $l$ and resolution $r$. In the case of a novel class the proxy can be utilized directly as the weight filter. An average of all the masked pooling features for the $k$-shot samples provided in the support set is used.

In case of the previously learned classes, the convolutional layer weights in our model can be updated with the newly imprinted weights for that class in an adaptive scheme. A running average is used to update the weights following equation 2 with the update rate $\alpha$. Figure 1 shows the adaptive masked weight imprinting scheme. Masked weight imprinting is performed on multiple resolution levels in order to improve the segmentation accuracy.

Our proposed online adaptation method can be followed by back-propagation, instead of performing back-propagation on randomly generated weights. Starting with random weights could cause over-fitting when fine-tuning with few labelled samples. However, with the proposed adaptation method it can be interleaved with back-propagation and ensures that the few labelled samples are properly utilized to change the convolutional filters.

$$\hat{W}_l^r = \alpha \hat{P}_l^r + (1 - \alpha) W_l^r \tag{2}$$

## 3 EXPERIMENTAL RESULTS

The setup for pretraining the models to be tested on PASCAL-$5^i$ is detailed. The base network is trained using RMSProp Hinton with learning rate $10^{-6}$, and L2 regularization with a factor of $5 \times 10^{-4}$ on the 15 classes outside of the current fold. In the few-shot evaluation 1000 samples are used similar to OSLSM setup Shaban et al. (2017). The alpha parameter used for adapting the previously learned weights is 0.26 for the 1 shot and 0.20 for the 5 shot. Table 1 and Table 2 show the results using the evaluation method from Co-FCN Rakelly et al. (2018) and the method in OSLSM Shaban et al. (2017) respectively on PASCAL-$5^i$. Our method is compared to the state of the art and the baseline methods for few-shot segmentation. Our method outperforms OSLSM and Co-FCN in the 5-shot case. Additionally, our method does not need to train an extra branch for predicting the parameters. Figure 2 shows the qualitative results on PASCAL-$5^i$ which shows both the support set image-label pair, and our predicted segmentation for the query image.

We conducted further experiments on iPASCAL, where triplets for the task, the corresponding images and semantic labels are provided. Semantic labels include the new classes in the current and previous encountered tasks. Figure 3 shows the comparison between naive fine-tuning from random weights against our proposed adaptive masked weight imprinting without any fine-tuning operations in terms of mIoU. It shows that masked imprinting provides better mIoU in comparison to fine-tuning that will lead to over-fitting. Fine-tuning was conducted using RMSProp with learning rate $10^{-10}$. Fine-tuning is performed only to the last layers responsible for pixel-wise classification, while the feature extraction weights for VGG16 are fixed. It is worth noting that the current evaluation setting is a $n$-way 1-shot, where $n$ increases with 2 additional classes with each encountered task resulting in 10-way 1-shot evaluation in the last task.

Table 1: Quantitative results for 1-way 1-shot and 5-shot segmentation on PASCAL-$5^i$ following evaluation in Rakelly et al. (2018). FT: Fine-tuning. PL+SEG by Dong & Xing (2018).

| Method | 1-Shot | 5-Shot |
|---|---|---|
| OSLSM | 55.2 | - |
| co-FCN | 60.1 | 60.8 |
| PL+SEG | **61.2** | **62.3** |
| Imp. (Ours) | 59.2 | 61.9 |
| Imp. + FT (Ours) | **61.6** | **65.9** |

Table 2: Quantitative results for 1-way 5-shot segmentation on PASCAL-$5^i$ dataset using evaluation by Shaban et al. (2017). FT(2): fine-tuning with 2 iterations.

| | LogReg | OSLSM | co-FCN | Imp. (Ours) | Imp. + FT(2) (Ours) |
|---|---|---|---|---|---|
| Fold 0 | 35.9 | 35.9 | 37.5 | 45.3 | **46.5** |
| Fold 1 | 51.6 | 58.1 | 50.0 | 51.4 | **53.9** |
| Fold 2 | 44.5 | 42.7 | 44.1 | 44.9 | **46.5** |
| Fold 3 | 25.6 | 39.1 | 33.9 | 39.5 | **41.2** |
| Mean | 39.3 | 43.9 | 41.4 | 45.3 | **47.0** |

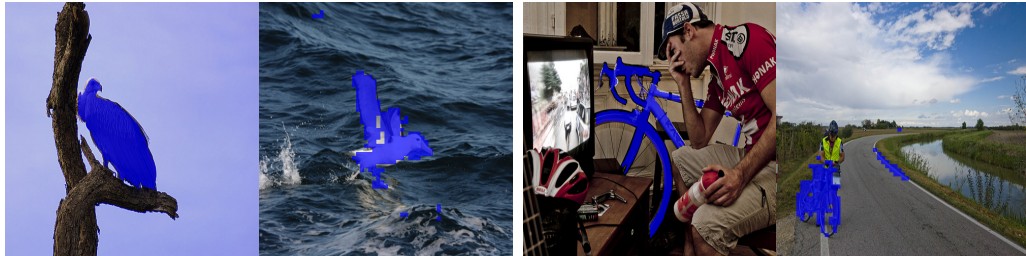

Figure 2: Qualitative evaluation on PASCAL-$5^i$. The support set and our proposed method prediction on the query image are shown in pairs for the 1-way 1-shot setting.

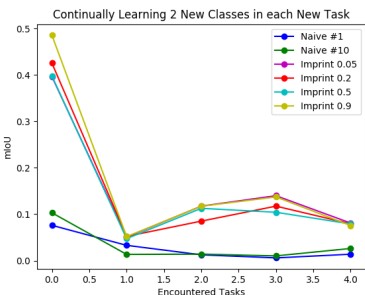

Figure 3: N-way 1-shot evaluation using the setup proposed in iPASCAL. Naive #1 denotes fine-tuning with 1 iteration per sample, while #10 uses 10 iterations. The Imprint method is utilized with different alpha parameters 0.05, 0.2, 0.5, 0.9.

## 4 CONCLUSION

In this paper we proposed a novel approach for few-shot semantic segmentation using an adaptive masked imprinting scheme. Our proposed method outperforms the state of the art few-shot segmentation methods in the 5-shot setting, while it alleviates the need for training a second branch as the previous literature. We proposed a novel setup iPASCAL to evaluate the effectiveness of our adaptation method.

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
