# OpenReview forum: "Adaptive Masked Weight Imprinting for Few-Shot Segmentation"
_ICLR.cc/2019/Workshop/LLD — LLD 2019_

### Official Review · AnonReviewer2 · 2019-04-08
**Authors propose a few-shot semantic segmentation using an adaptive masked imprinting scheme on the Pascal 5^i dataset**

**Rating:** 3
**Confidence:** 2

**Review:**

Authors use an imagenet pretrained VGG-16 o perform few-shot semantic segmentation scheme where the goal is to adaptively update the weights to perform few shot semantic segmentation with task, image, label pairs. Once a base classifier is trained, the embedding vectors of new of the support set are used to imprint weights (learn a metric w.r.t e novel class as proxies) for new classes in the extended classifier.

- What is Wnew, Wold and Wimprinted, and more complete definition would be useful since this is the key contribution.
- Results seem quite interesting, though understand why an adaptive update for the weights in a continous stream setup, would help complete the understanding of the paper.
- If the continous stream is a video where the frames are correlated temporally would this approach help ?

---

### Official Review · AnonReviewer1 · 2019-04-08
**Important problem and good results but the text needs significant corrections and improvements.**

**Rating:** 2
**Confidence:** 1

**Review:**

This paper proposed a method for training a convolutional neural network (CNN) from few examples (few-shot learning) for the task of semantic segmentation. The proposed technique allows use to incrementally update the weights of the CNN when encountering examples from classes the networks has not seen before. The paper builds on the idea of weight imprinting, introducing an adaptive weight imprinting scheme that enables updating the weights of previously learned classes. The results on the PASCAL-5^i dataset look convincing, however the text lacks clarity and needs to be proofread and corrected.  Specifically, I have the following questions/comments:

    Multiple grammatical and syntactical  errors make the text hard to follow, e.g.,  the last sentence of the first paragraph of the introduction does not make sense, and citations need fixing.

    "Masked weight imprinting is performed on multiple resolution levels in order to improve the segmentation accuracy" --> the multiscale nature of the algorithm is not captured by eq. 2. It is not clear how weight imprinting is performed at multiple resolutions.

    What does "interleaved with backpropagation" mean? This should be clarified in the text.

---

### Decision · Program_Chairs · 2019-04-12
**Acceptance Decision**

**Decision:**

Accept

**Comment:**

Interesting idea and good results. The paper needs some thorough writing update.